# Effects of SARS-CoV-2 Pandemic on the Mental Health of Spanish Ob-Gyn Specialists—A Nationwide Study

**DOI:** 10.3390/jcm10245899

**Published:** 2021-12-15

**Authors:** Ernesto González-Mesa, Jesus Salvador Jiménez-López, Marta Blasco-Alonso, Jose Ramon Anderica-Herrero, Daniel Lubián-López

**Affiliations:** 1Department of Obstetrics and Gynecology, School of Medicine, University of Malaga, 29071 Malaga, Spain; egonzalezmesa@gmail.com (E.G.-M.); martablascoalonso@gmail.com (M.B.-A.); 2Department of Obstetrics and Gynecology, SSPA Regional University Hospital of Málaga, 29011 Malaga, Spain; jranderica@hotmail.com; 3Spanish Section of Psychosocial Obstetrics and Gynecology, 28036 Madrid, Spain; dmlulo@gmail.com; 4Department of Obstetrics and Gynecology, University Hospital of Jerez de la Frontera, 11407 Cadiz, Spain

**Keywords:** SARS-CoV-2, mental health, obstetrics and gynecology

## Abstract

It has been reported that frontline health professionals have suffered from an increased rate of mental disorders as a consequence of the SARS-CoV-2 pandemic. Objective: This study aims to evaluate the impact of the pandemic on the mental health of Spanish specialists in the obstetrics and gynecology fields after the fifth Spanish COVID-19 wave. Methods: Psychometric screening of the professionals was performed using a nationwide online survey designed by the Psychosocial Obstetrics and Gynecology taskforce. Post-traumatic stress disorder (PTSD), depression, and general anxiety disorder were screened. Results: We found high scores for PTSD, depression, and generalized anxiety disorder. Up to 12.4% of specialists met the PTSD criteria, 12.8% showed complex PTSD, 37% showed depression, and up to 37% had a positive screening for generalized anxiety disorder. Our results showed that the pandemic has particularly affected the group of female Ob-Gyn specialists, with significant higher scores in depression and anxiety scales. Conclusions: As a lesson learned for the future in case of new health emergencies, it would be very important to provide professionals with adequate psychological support, ensuring enough human and material resources, flexible work shifts, and facilities to reconcile work and family-life, especially in the case of female specialists.

## 1. Introduction

After the COVID-19 outbreak was declared as a public health emergency of international concern in January 2020 [1], five pandemic periods have been declared in Spain [2]. According to official data, between 22 June 2020 and 3 November 2021, more than five million cases have been registered in Spain. A total of 323,693 required admission to a hospital, 31,983 were admitted to intensive care units, and 57,589 people died as a result of the infection. Among the infected subjects, 61,272 were health professionals [2].

As one of the collateral effects of the pandemic, the mental health of the general population has been affected; specifically, working in essential activities and being in contact with infected patients have been reported as two main risk factors of psychological distress [3]. As especially vulnerable populations [4], health care workers, who are both essential workers and exposed to SARS-CoV-2 infected individuals, have been described as suffering from mental disorders as a consequence of the pandemic [5]. This observation is more intense in the case of frontline health professionals working in emergency conditions in which the threat of transmission is higher due to the requirement of working with sick people with high viral loads and, typically, using sub-optimal personal protection equipment [4,6]. During the pandemic, specialists in obstetrics and gynecology (Ob-Gyn) have endured the crisis with an overload of emergency conditions, including emergency gynecological surgery, gynecological oncology, and perinatal care; furthermore, they often must provide support in non-gynecological areas, conducting tasks that are not possible to manage remotely with telemedicine [7]. In addition to the usual stress of the specialty, these professionals have been fighting with the emotional overload due to the pandemic, heavy working shifts, concerns of being infected or even infecting their relatives, lack of protective equipment, and feelings of ineffectiveness due to restricting the accompaniment of patients—especially women in labor—which could affect their mental health.

This study was designed to evaluate the impact of the pandemic on the mental health of Spanish specialists in obstetrics and gynecology, considering the epidemiological moment in which the incidence rates seem to have decreased after the fifth pandemic wave.

Most of the Spanish Obstetrics and Gynecology hospital services screened for SARS-CoV-2 infection upon admission of patients, and before surgical interventions [8] along all pandemic waves. This allowed to have a fairly accurate idea of the incidence of the infection in admitted women. Data from the Spanish registry of COVID-19 in asymptomatic pregnant women showed that more than 2% of asymptomatic women admitted in labor had a positive test [9], and pregnant women with COVID-19 were reported to be at a higher risk of caesarean section and preterm labor [10], what challenged maternity services. On the other hand, the incidence of preoperative positive SARS-CoV-2 tests in asymptomatic women was under 0.1% [11]. Fortunately, the incidence and severity of COVID-19 in obstetric population remained relatively low during the first pandemic waves, but management protocols were adapted to prevent the spread of the infection.

The fifth pandemic wave occurred in Spain during July and August 2021, reaching a fourteen-day cumulative incidence rate of 321.1 cases per 100,000 inhabitants. The vaccination strategy in our country [12] had prioritized the use of the COVID-19 vaccine in older age groups and populations at special risk, so that although in September 2021 the proportion of vaccinated Spanish citizens was close to 70%, many young people had not received the vaccine yet. Specifically, the cumulative incidence rate in the age group of 30 to 40 years old, which included most of the women at childbearing age, reached 401 cases per 100,000 [2]. This scenario has been especially challenging for specialists in obstetrics and gynecology, due to the high prevalence of the disease in young age groups and pregnant women.

## 2. Methods

### 2.1. The Survey

The Spanish Society of Obstetrics and Gynecology (SEGO) is the main national professional membership organization for obstetricians and gynecologists in Spain [13]. With the approaching end to the pandemic-related restrictions, SEGO has been preparing for its first post-pandemic face-to-face meeting in which the effects of the pandemic on the physical and mental health of Ob-Gyn specialists were expected to be discussed. For their mental health assessment, a psychometric screening of these professionals was designed, using a nationwide online survey designed by the Psychosocial Obstetrics and Gynecology taskforce. The survey was conducted in September of 2021, after ethical permissions were approved by the local committee (comité de ética de la investigación de la provincial de Malaga)—Research ethics committee of the province of Malaga (Approval Code: 1194/2021. Approval Date: 14 February 2021). An online invitation was sent by email to all participants, with a link to the questionnaire. Three appeals were made at 7-day intervals. Around 800 SEGO members had confirmed their registration for the meeting at the time we ran the survey. We finally received feedback from 220 specialists, representing 27.5% of the registered partners at the time that the survey was sent.

### 2.2. Instruments

The questionnaire consisted of a brief explanation of the reasons why the research was being conducted, the informed consent document, some sociodemographic questions (e.g., gender, age, city of residence, and job title), and a series of questions regarding their activity during the past SARS-CoV-2 pandemic waves (i.e., if they had been working as frontline professionals, attending to infected women either in site consultations, in surgical areas, in delivery rooms, or in COVID-19-specific hospital areas). It also included some questions regarding their pre-pandemic health status and possible changes in recent months. Specifically, we asked about SARS-CoV-2 infection and severity, as well as possible pre- or post-pandemic clinical diagnosis of high blood pressure, diabetes mellitus, menstrual disorders, thyroid disturbances, insomnia, depression, anxiety disorders, phobias, and the use of psychotropic drugs (e.g., antidepressants, selective serotonin reuptake inhibitors, and benzodiazepines). Finally, the questionnaire included three psychometric tools to screen for post-traumatic stress disorder (PTSD), depression, and generalized anxiety disorder.

To assess PTSD, we used the International Trauma Questionnaire (ITQ). This scale is a self-reported questionnaire, based on the International Classification of Diseases (11th revision) criteria for PTSD and Complex PTSD (CPTSD) [14]. PTSD and CPTSD diagnosis are based on the exposure to a potentially traumatic event, and while PTSD is defined by the presence of re-experiencing symptoms (Re), avoidance (Av) of stimuli associated with the trauma, and a sense of current threat (Th), the diagnosis of CPTSD includes the existence of self-organized disturbances, like affect dysregulation (AD), negative self-concept (NSC), and disturbances in relationships (DR) related to the trauma. It includes 22 items assessing how often participants experience the six symptoms of PTSD (two from each of the three sub-scales) within the last month. The questionnaire also includes three functional impairment items related to these sub-scales, on a 5-point Likert scale that ranges from 0 (‘not at all’) to 4 (‘extremely’). To reach a diagnosis of PTSD, at least one of two symptoms from each PTSD symptom sub-scale (score ≥2), and at least one of the functional impairment items are needed. A diagnosis of CPTSD requires the existence of one symptom from each of the three PTSD symptom groups (re-experiencing, avoidance, and sense of current threat), as well as one symptom from each of the Disturbances in Self-Organization (DSO) previously mentioned groups: affective dysregulation, negative self-concept, and disturbances in relationships. This scale has been previously translated into Spanish and validated [15].

Depression was screened using the Patient Health Questionnaire-9 (PHQ-9) [16]. This is a nine-item self-reported questionnaire based on the Diagnostic and Statistical Manual of Mental Disorders (fifth edition) criteria for depression. It comprises a four-point Likert scale that ranges from 0 (‘not at all’) to 3 (‘very much’). Final scores are the sum of all item scores (0–27); a score of 10 or higher indicates moderate depression. The scale has been translated into Spanish and validated in a sample of Spanish depressive patients [17], and scores ≥10 resulted in 84% sensitivity and 92% specificity for the diagnosis of major depression

We screened generalized anxiety disorder with the GAD-7 scale [18] (GAD-7). This tool is a seven-item self-reported questionnaire developed to assess the severity of anxiety symptoms. The items are scored on a four-point Likert-type scale in reference to the past two weeks. Participants report how much each symptom affected them, scoring from 0 (‘not at all’) to 3 (‘more than half the days’). Final scores are the sum of all item scores (0–21). The Spanish version of the scale has been previously validated, where the use of a score of 10 as a cut-off point resulted in a sensitivity of 86.8% and a specificity of 93.4% [19].

The research was carried out in accordance with the ethical criteria of the Helsinki Declaration, preserving the rights and confidentiality of all patients.

### 2.3. Population

A final sample of 220 Ob-Gyn specialists participated in the survey and completed the questionnaire. Most of the respondents were women (68.1%). The mean age of the participants was 46 years old, with significant differences according to gender (Chi Sq = 48.2; *p* < 0.001): in the group of female specialists, the mean age was 37 years old while, in the group of male Ob-Gyn doctors, the mean age reached 56 years old. Main sociodemographic features are shown in Table 1.

### 2.4. Statistics

First, we carried out validation of the three psychometric tools used in our sample. Then, we studied the score distributions and performed bivariate and multivariate analyses. We used ANOVA and *t*-test for bivariate quantitative/qualitative comparisons, and Chi-square test (Chi sq) for bivariate qualitative/qualitative comparisons. For multivariate analyses, we used multiple linear regression to build predictive models. Stepwise regression procedures were performed, initially including all sociodemographic and health-related variables that showed significant associations in the bivariate analysis. We also studied the correlations between the ITQ, PHQ-9, and GAD-7 scores, through calculating Pearson’s coefficients and their significance levels.

For the validation of the questionnaires, we used the statistical analysis of Cronbach’s alpha coefficient to measure the internal consistency of the scores. The Bartlett´s test of sphericity, and the Kaiser–Meyer–Olkin (KMO) test were used to assess the acceptability of the factor analysis. Exploratory factor analysis was conducted using the analysis of the main components of the questionnaires, and the Varimax method was used to identify latent factors. Eigenvalues (>1) and scree plots were examined to determine the number of factors for the scales. Confirmatory factor analysis was carried out using structural equation analyses that was performed considering the maximum likelihood estimator. The goodness of fit was analyzed using the standardized root mean square (SRMS), Tucker–Lewis index, comparative fit index (CFI), and root mean square error of approximation (RMSEA). The acceptable goodness of fit was defined by SRMR or RMSEA values ≤0.08, Tucker–Lewis index ≥0.95, and CFI ≥ 0.90 [20,21]. The statistical analysis was carried out using the Stata version 14.0 (Stata Corp., TX, LP, USA) software, and SPSS version 25.0 (IBM SPSS Statistics for Windows, Version 25.0, released 2018, IBM Corp., Armonk, NY, USA).

## 3. Results

Table 2 shows the information regarding changes in health status over time; that is, comparing their status before the first pandemic outbreak to that at the moment of completing the survey.

We found that 25% of participants had been infected with SARS-CoV-2 at any time during the pandemic, mostly with asymptomatic or mild clinical presentations. Most of the respondents acted as frontline workers either in surgical areas, delivery rooms, or consultation areas, with some even in specific internal medicine COVID-19 areas. Table 3 shows information regarding specialist activity during the pandemic, disaggregated by gender. Table 3 shows clinical variables of the participants.

We observed high scores in the three psychometric scales. Table 4 shows the main statistics for these instruments and their correlation coefficients. The scores showed that 12.4% of the specialists presented PTSD, 37% presented depressive symptoms, and 37% presented general anxiety disorder.

### 3.1. Post-Traumatic Stress Disorder (ITQ)

The Cronbach´s alpha coefficient for ITQ was 0.94. Barlett and Kaiser–Meyer–Olkin´s tests revealed the adequacy of the samples to perform an exploratory factor analysis (KMO = 0.92; Chi Sq = 2571.3, 153 df; *p* < 0.0001). The expected three factors were found to explain 68.17% of the global variance in the distribution of PTSD and CPTSD scores (re-experiencing, avoidance, and sense of current threat for PTSD; and affective dysregulation, negative self-concept, and disturbances in relationships for CPTSD).

We found that 12.4% of the participants scored over the cut-off for PTSD, with 20.7% reporting re-experiencing in the here and now, 30% avoidance, 47.5% sense of current threat, and 43.8% functional impairment. We also found that 12.0% of the participants scored over the cut-off for CPTSD, with 53.9% presenting affective dysregulation, 22.6% negative self-concept, 41% disturbances in relationships, and 41.5% functional impairment due to disturbances in self-regulation.

PTSD was significantly more prevalent (21.8%) in specialists over 55 years old (Chi Sq = 9.3; *p* < 0.05). Furthermore, while 15.3% of female doctors scored positively for PTSD, only 6% of male specialists did (Chi Sq = 3.72; *p* < 0.05). Multiple regression showed that the best predictor for the PTSD variable was gender (R^2^ = 0.045; Standardized beta coefficient = 0.21; *t*-value = 3.3; *p* < 0.002). SEM analyses showed that the best-fitting model included age and frontline work in surgical areas and delivery rooms (Figure 1).

### 3.2. Depression (PHQ-9)

For the PHQ-9, we determined a Cronbach’s alpha score of 0.93. Barlett and Kaiser–Meyer–Olkin´s tests revealed adequacy of the samples to perform an exploratory factor analysis (KMO = 0.94; Chi-Square = 1363.2, 36 df; *p* < 0.0001). One single factor explaining 65.28% of the global variance was found.

PHQ-9 scores were over the cut-off point for depression in 37% of the participants, with a 10.0% of specialists scoring over the cut-off severe depression. Scores significantly differed according to the geographical area, with Madrid being the region where the scores were the highest (Mean = 12.5; SD = 7.4). We found higher mean scores in the group of female specialists (mean value = 9.47, SD = 6.5 in women; mean = 7.49, SD = 8.0 in men; *t*-value −1.91; *p* < 0.05). Specifically, women scored significantly higher for severe (18.1% of women vs. 7.5% of men) and very severe (6% of women vs. 3% of men) depression (Chi sq = 21.3; *p* < 0.001). We also found higher PHQ-9 scores in specialists that had attended to COVID-19 patients in surgical areas (mean score = 9.9, SD = 7.3 vs. mean = 7.58, SD = 6.4; F = 6.04; *p* < 0.05). Women with higher PHQ-9 scores also reported a higher incidence of menstrual disorders (F = 4.5; *p* < 0.001). On the other hand, we did not find any difference in the scores according to the age of the specialists. PHQ-9 scores did not differ among the specialists, according to either their own or relative’s SARS-CoV-2 infection, severity of COVID-19 infection, or vaccination status.

According to multivariate regression analysis, the best predictor variables for PHQ-9 scores were previous depression diagnosis and working in surgical areas during the pandemic (Table 5).

The final best-fitting model and goodness of fit parameters are shown in Figure 2.

### 3.3. General Anxiety Disorder (GAD-7)

For the GAD-7, we found a Cronbach’s alpha score of 0.94. Bartelett and Kaiser–Meyer–Olkin´s tests revealed the adequacy of the samples to perform an exploratory factor analysis (KMO = 0.94; Chi-Square 1283.2, 21 df; *p* < 0.0001). One single factor explaining 75.1% of the global variance was found. Anxiety scores were over the cut-off point in 37% of the specialists. Again, we found significantly higher scores in female specialists (mean values reaching 9.45 in women (SD = 5.7) vs. 6.24 (6.3) in men; F = 13.25; *p* < 0.0001) and in the group of frontline workers (8.94 in specialists that had treated COVID-19 patients [8.9] vs. 6.35 [5.6] in gynecologists that had not; F = 5.2; *p* < 0.023). We found that anxiety scores were specifically higher in physicians that had been working with patients either in surgical areas (mean value = 9.53; SD = 6.1; F = 7.29; *p* < 0.001) or in delivery rooms (mean value = 9.32; SD = 6.1; F = 5.44; *p* < 0.05), when compared to professionals attending in-site consultations (mean value = 8.8; SD = 6.2). Multivariate regression models showed that the best predictor variables were gender and working in surgical areas during the pandemic (Table 6). SEM analysis showed that, when including the variable ‘delivery room specialist’, the fitness of the model improved. Goodness-of-fit parameters are provided in Figure 3.

## 4. Discussion

This research was designed to determine the impact of the pandemic on the mental health of obstetricians and gynecologists. We verified the existence of a high burden on the mental health of professionals, as we observed high scores for PTSD, depression, and generalized anxiety disorder scales. Up to 12.4% of specialists met PTSD criteria, 12.8% showed complex PTSD, 37% showed depression, and 37% had a positive screening for generalized anxiety disorder. The stress caused by the progress of the pandemic and the thousands of associated deaths, the absence of effective treatment, the saturation of hospitals [2], and long work shifts with potential exposure to infection have influenced the mental health of professionals [22,23]. During the pandemic, specialists have participated in virus containment strategies [24], proposing treatment protocols and measures to prevent the spread of the virus, on many occasions establishing strict isolation measures for patients (e.g., specific areas with isolation rooms, visiting restrictions). Unfortunately, many of them have been infected by SARS-CoV-2, suffering the harsh isolation measures that they had proposed as patients [25]. They have been exposed, as doctors, to sick patients many times, making them responsible for virus prevention measures (and their collateral consequences), while being susceptible to becoming victims of the disease and the isolation measures themselves. Professionals have alternatively been the figure and the ground of the pandemic stage in the hospitals, which has inevitably overloaded their emotional control mechanisms.

The mental health of women has been reported to be especially vulnerable to the pandemic [26,27,28,29]. Our results showed that the pandemic has particularly affected the group of female Ob-Gyn specialists. The evolution of the pandemic forced absolute confinement during the first few months, in addition to the cessation of face-to-face school classes for children, among other restrictions. Although teleworking has been possible for some jobs, health professionals in general, and specifically obstetricians and gynecologists, have been working their usual shifts (when they were not overtime), with the additional responsibility of being family caregiver [30], caring for school-age children at home in many cases, helping in their virtual schooling and, ultimately, supporting the family. This, together with the task of working on the front line of the fight against the infection, has generated an emotional overload that we can now perceive.

On the other hand, an association between SARS-CoV-2 affectation and menstrual alterations has been reported [31,32]. In this regard, it should be noted that, in our sample, we observed that an important group of women experienced the debut of menstrual alterations during the pandemic. Women with menstrual disturbances have shown a higher proportion of depressive and anxiety symptoms, which have been also shown to be factors modifying the menstrual cycle [33]. In any case, these findings point to the group of women as being particularly vulnerable.

Furthermore, our results clearly indicate that the emotional impact is directly related to acting as frontline professionals; especially in surgical settings in which the need to treat non-delayed pathologies renders professionals more exposed and vulnerable. Front line gynecological surgeons had to face emergency interventions in patients affected with COVID-19, and delivery-room professionals had to care for women in labor regardless of their COVID-19 status.

Mental illness produces great personal suffering, but has also been associated with loss of concentration, increased medical errors and, ultimately, worse care for patients. The negative effects of the pandemic on the mental health of the general population [34,35,36,37], and specifically that of health care workers, are not surprising [38,39,40,41]. They had been previously communicated after previous coronavirus pandemics in the beginnings of the 20th century, and some preventive measures have even been suggested [42,43]; however, little of this seems to have been considered. In the future, human and material resources, flexible work shifts, and facilities to reconcile work and family life—especially for the group of women specialists—must be ensured.

Some limitations of this study need to be considered. First, the psychometric instruments that were used, although previously validated as having good sensitivity and specificity profiles, were based on self-reported questionnaires. Second, the study was cross-sectional, and causal inferences could not be determined. Moreover, we did not have accurate information about the pre-pandemic mental health status of these physicians. Furthermore, although the sample of the study is representative of the physicians that were registered for the SEGO meeting at the time of sending the survey, the sample was small and may not be fully representative of the population of Spanish Ob-Gyn specialists. Finally, when asking the participants about their pre-pandemic health status, despite all the questions were objective and concise, there might have been some recall bias in the answers. However, this is the first study aimed at providing specific information about the mental health of Spanish Ob-Gyn specialists during the pandemic. Some previous studies [16,17,18,19] designed to assess the impact of the COVID-19 pandemic on mental health among health care workers have included obstetricians and gynecologists, but the samples were significantly smaller. Furthermore, this work provides important information to be considered in the management of future pandemics.

## 5. Conclusions

The effects of the pandemic on the mental health of specialists are evident, with an increase in cases of depression and generalized anxiety, and a significant rate of post-traumatic stress. The group of female Ob-Gyn specialists was the one in which the greatest affectation was observed. Furthermore, professionals working in surgical and perinatal areas were significantly affected. Currently, it is necessary to identify the professionals who need help when suffering from mental sequelae, offering them appropriate psychological support. In the future, it will be necessary to establish measures to prevent the impact of possible new pandemics on the mental health of health care professionals.

## Figures and Tables

**Figure 1 jcm-10-05899-f001:**
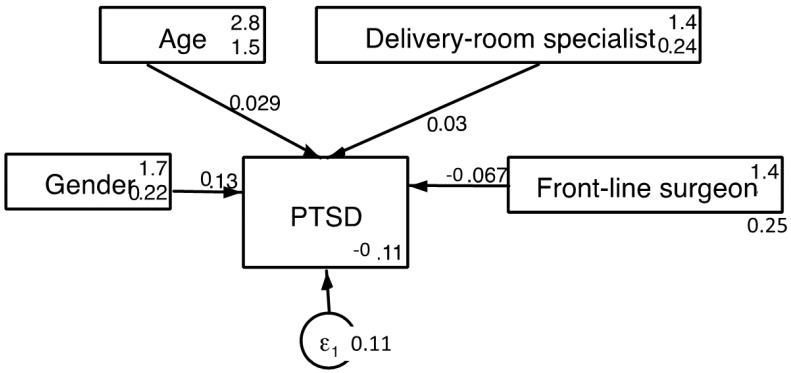
Predictive model for ITQ scores; PTSD, Post-traumatic stress disorder; Chi sq (baseline vs. saturated) = 6.588; *p* < 0.159; Population error RMSEA = 0.001 (*p* < 0.05); Akaike’s information criterion = 1462.263; Bayesian information criterion = 1482.458; Comparative fit index = 1; Tucker–Lewis index = 1; Standardized root mean squared residual = 0.001; Coefficient of determination = 0.030.

**Figure 2 jcm-10-05899-f002:**
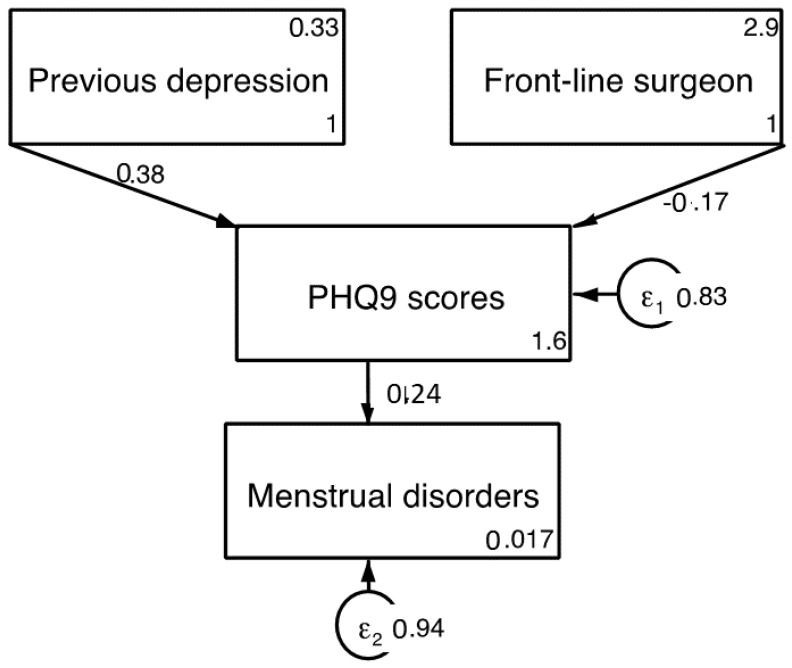
Predictive model for PHQ-9 scores. Chi sq (model vs. saturated) = 4.289; *p* < 0.117; Population error RMSEA = 0.073 (*p* < 0.05); Akaike’s information criterion = 2528; Bayesian information criterion = 2552.213; Comparative fit index = 0.957; Tucker–Lewis index = 0.892; Standardized root mean squared residual = 0.036; Coefficient of determination = 0.173.

**Figure 3 jcm-10-05899-f003:**
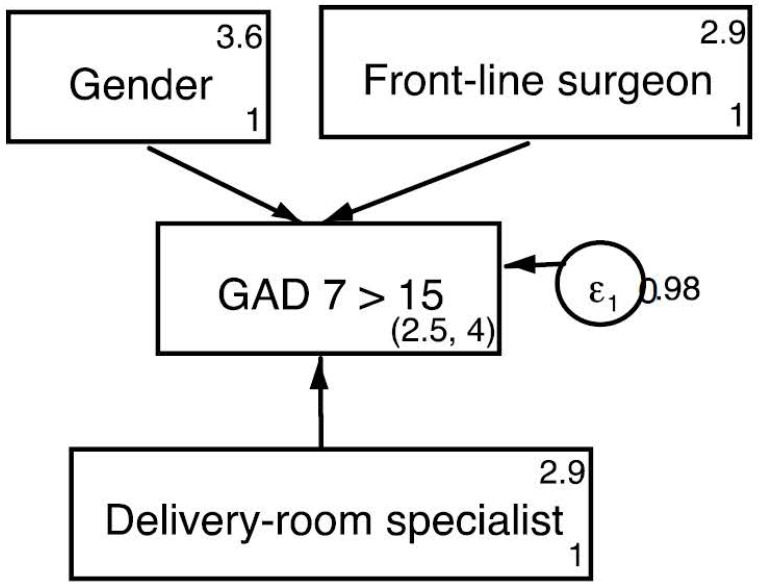
Predictive model for positive GAD screening. Chi sq (baseline vs. saturated) = 3.853; *p* < 0.278; Population error RMSEA = 0.001 (*p* < 0.05); Akaike’s information criterion = 875.80.; Bayesian information criterion = 892.64; Comparative fit index = 1; Tucker–Lewis index = 1; Standardized root mean squared residual = 0.001; Coefficient of determination = 0.018.

**Table 1 jcm-10-05899-t001:** Sociodemographic variables.

Variable	*n* (%)
Gender	
Women	150 (68.1%)
Men	70 (31.9%)
Age	
26–35	41 (18.9%)
36–45	53 (24.4%)
46–55	52 (24.0%)
56–65	55 (25.3%)
Older than 65	16 (7.4%)
Region	
Región de Murcia	12 (5.5%)
Principado de Asturias	5 (2.3%)
País Vasco	6 (2.8%)
La Rioja	1 (0.5%)
Islas Baleares	4 (1.8%)
Galicia	10 (4.6%)
Extremadura	5 (2.3%)
Comunidad Valenciana	18 (8.3%)
Comunidad de Navarra	2 (0.9%)
Comunidad de Madrid	44 (20.3%)
Ceuta	1 (0.5%)
Cataluña	15 (6.9%)
Castilla-La Mancha	15 (6.9%)
Castilla y León	10 (4.6%)
Cantabria	2 (0.9%)
Canarias	6 (2.8%)
Aragón	6 (2.8%)
Andalucía	55 (25.3%)

**Table 2 jcm-10-05899-t002:** Clinical variables.

Variable	n (%)	Variable	n (%)
Menstrual disorders *	(only pre-menopausal women)		
No	129 (89.9%)	Depression *	193 (87.7%)
Before pandemic	6 (4.6%)	No	8 (3.6%)
Now	15 (11.6%)	Before pandemic	18 (8.1%)
	χ^2^ = 4.05; *p < 0.05*	Now	χ^2^ = 4.087; *p ≥ 0.05*
Diabetes Mellitus		SSRI	
No	210 (96.8%)	No	204 (92.7%)
Before pandemic	3 (1.8%)	Before pandemic	7 (3.1%)
Now	3 (1.8%)	Now	7 (3.1%)
High blood pressure		Insomnia	
No	196 (90.3%)	No	156 (71.9%)
Before pandemic	15 (3.7%)	Before pandemic	33 (15.2%)
Now	17 (5.2%)	Now	37 (17.0%)
Thyroid disorders		Benzodiazepines *	
No	206 (94.9%)	No	203 (92.2%)
Before pandemic	6 (2.5%)	Before pandemic	3 (1.3%)
Now	6 (2.5%)	Now	13 (5.9%)
			χ^2^ * = 10.62;* *p < 0.0001*
Anxiety *		Sleeping pills	
No		No	191 (86.8%)
Before pandemic	187 (85.0%)	Before pandemic	13 (5.9%)
Now	6 (2.7%)	Now	16 (7.2%)
	30 (13.6%)		
	χ^2^ = 16.42; *p* < 0.0001		
Phobias			
No	205 (93.1%)		
Before pandemic	4 (1.8%)		
Now	9 (4.1%)		

SSRI, Serotonine selective reuptake inhibitors; (*) significant differences.

**Table 3 jcm-10-05899-t003:** Activity of the participants during the pandemic.

	Overall	Women	Men
Infected by SARS-CoV-2			
Yes	54 (24.9%)	35 (23.5%)	19 (28.4%)
No	162 (74.7%)	114 (76.5%)	48 (71.6%)
COVID-19 severity			
Asymptomatic	7 (2.8%)	3 (8.5%)	4 (21.0%)
Mild	35 (16.1%)	23 (65.5%)	12 (63.1%)
Pneumonia	9 (4.1%)	7 (20.0%)	2 (10.5%)
Intensive Care	3 (1.4%)	2 (5.7%)	1 (5.2%)
Severely ill/deceased relatives			
Yes	49 (24.9%)	31 (20.81%)	18 (26.9%)
No	167 (74.7%)	118 (79.2%)	49 (73.1%)
Frontline workers			
Yes	181 (83.4%)	127 (85.8%)	54 (80.6%)
No	34 (15.7%)	21 (14.2%)	13 (19.4%)
Treated COVID-19 patients			
Birth/Delivery room *	130 (59.9%)	97 (65.5%)	33 (49.3%)
Chi Sq 5.11; *p* < 0.05			
Surgical areas	120 (55.3%)	88 (59.5%)	32 (47.8%)
Consultations	170 (78.3%)	121 (81.8%)	49 (73.1%)
Non-Ob-Gyn areas	57 (26.3%)	40 (27%)	17 (25.4%)
Needed Personal Protective Equipment *			
Yes			
No	142 (79.3%)	125 (84.5%)	47 (70.1%)
Chi Sq 5.093; *p* < 0.005	43 (19.8%)	23 (15.5%)	20 (29.9%)
Vaccinated against SARS-CoV-2			
Yes	217 (99%)	147 (99.3%)	65 (97%)
No	3 (1%)	1 (0.7%)	2 (3%)
Sick leave or quarantined			
Yes	66 (30.4%)	52 (35.1%)	14 (20.9%)
No	149 (68.7%)	96 (64.9%)	53 (79.1%)
Chi Sq 4.39; *p* < 0.005			

* significant differences.

**Table 4 jcm-10-05899-t004:** Statistics for the psychometric instruments.

	ITQ	GAD-7	PHQ9
Mean	5.26	8.46	8.86
Median	4	8	8
Std. Deviation	5.129	6.155	7.069
Percentiles 25th	1	3	3
50th	4	8	8
75th	8	13	14
Pearson’s *r*			
ITQ	1	0.63 *	0.61 *
GAD-7	0.63 *	1	0.84 *
PHQ-9	0.61 *	0.84 *	1

Significant correlation coefficients (*p* < 0.05) have been marked (*).

**Table 5 jcm-10-05899-t005:** Predictive model for PHQ-9 scores.

	Unstandardized B	Std. Error	Beta Coefficient	*t*-Value	*p*
(Constant)	11.52	1.357		8.49	0.00
Frontline surgical	−2.44	0.88	−0.172	−2.75	0.00
Previous depression	4.66	0.74	0.381	6.10	0.00

**Table 6 jcm-10-05899-t006:** Predictive model for GAD-7 scores.

	Unstandardized B	Std. Error	Beta Coefficient	*t*-Value	Sig.
(Constant)	6.058	2.018		8.97	0.00
Frontline surgical	−1.92	0.812	−0.15	−2.36	0.01
Gender	3.11	0.871	0.23	3.57	0.00

## Data Availability

Data are available under reasonable request to egonzalezmesa@gmail.com.

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
