# Peer review of "Effects of SARS-CoV-2 Pandemic on the Mental Health of Spanish Ob-Gyn Specialists—A Nationwide Study"

_jcm, 2021, doi:10.3390/jcm10245899_

Round 1
Reviewer 1 Report
Abstract:
Needs some rewriting to properly reflect the important findings and conclusions of the paper.
I suggest significantly reducing the first 2/3 (background and objectives) then extend the results a little and add a sentence or two on the important implications/conclusions you draw. In particular, I think the effect you find on women specialists, who have added burdens (eg home schooling children) is particularly important and you should reiterate your call for future pandemic responses to offer “appropriate psychological support” and ensure “human and material resources, flexible work shifts, and facilities to reconcile work and family life”
Intro
What is meant by “non-delayed conditions”? Do you mean emergency conditions? Define or use a standard term.
Expand the detail on the Spanish epidemiological situation to give the reader a better understanding. You note that the 5th wave saw a decrease in incidence, but then you note that there is “high prevalence of the disease in young age groups and pregnant women” – was this during the 5th wave only (why? Different variants?) or a feature of all waves?
Methods
Well described and seem appropriate and interesting. What about recall bias? You are asking participants to remember what their health was like two years ago before the first wave – how accurate/unbiased is this likely to be? You need a comment on this either here or in your limitations paragraph at the end of the discussion.
I cannot comment on the statistical methods.
Section 2.3 – why are you presenting results in the methods section? T1 and 2 and all accompanying text should be moved to the results section. All you need in the methods is a brief description of the population (perhaps with a short table showing the sociodemographic variables but not any of your clinical findings).
Table 1: alignment in last column of first row (menstrual disorders) is out so hard to read correctly. It would make more sense to present first the sociodem variables (eg in the top half of the table) then the clinical variables – at present they are mixed up.
Results
This should start with tables 1 and 2 giving an overview of the respondents’ background characteristics and reported disorders/ill-health before and after the pandemic.
3.1: several times you use “re-experencing” – I think you mean “re-experiencing”
Fig 1: define TEPT. Also explain E. I don’t understand why the multiple regression showed gender as the best predictor for PTSD but this doesn’t then appear in the best-fit SEM model – please explain.
Fig 2: this is not cited in your text – you need to refer to it and explain it. I don’t understand the presentation of the sequencing – you found that depression scores were higher in women who also reported having menstrual disorders, but the way you present the Fig makes it seem as though depression leads to menstrual disorders. This is not what you found. Reversing the arrow might help to clarify this.
Discussion
Currently the first sections of this are rather thinly referenced with wider literature and you have many unsupported statements, which (however true they may be) need to be referenced. For example, your research does not examine the wider causes of the rise in mental health disorders that you found, therefore you need references to support statements like “The stress caused by the progress of the pandemic and the thousands of associated deaths, the absence of effective treatment, the saturation of hospitals, and long work shifts with potential exposure to infection have damaged the mental health of professionals” and “specialists have participated in virus containment strategies, proposing treatment protocols and measures to prevent the spread of the virus, on many occasions establishing strict isolation measures for patients” and “obstetricians and gynecologists in particular, have been working their usual shifts (when they were not overtime), with the additional responsibility of caring for school-age children at home in many cases, helping in their virtual schooling and, ultimately, supporting the family.”
You could give more detail on the nature of the frontline work that is associated in your models with high mental health scores (across the various that you test for). What does frontline surgeon work in Spain involve, compared to delivery room specialists. Were there any differences between your male and female respondents in terms of where they worked?
Sensible implications and conclusions are drawn.
Author Response
Malaga 2021, December 10
Dear Editor of the Journal of Clinical Medicine:
We wish to thank the careful reading that the reviewers made of our article, as well as the comments and contributions, which undoubtedly have served to improve the quality of the study. We have done our best to incorporate the recommendations of the specialists who have reviewed the paper. We have considered point-by-point all the suggestions, including in the new document all the changes that are described below. We hope that the manuscript could be now accepted for publication in the journal.
On behalf of the authors,
Dr Jesús S Jiménez-López
Response to the Reviewers
REVIEWER 1
Abstract:
I suggest significantly reducing the first 2/3 (background and objectives) then extend the results a little and add a sentence or two on the important implications/conclusions you draw. In particular, I think the effect you find on women specialists, who have added burdens (eg home schooling children) is particularly important and you should reiterate your call for future pandemic responses to offer “appropriate psychological support” and ensure “human and material resources, flexible work shifts, and facilities to reconcile work and family life”
We have changed the abstract following your suggestions. We have highlighted the higher impact on women, and the call for future pandemic
Intro
What is meant by “non-delayed conditions”? Do you mean emergency conditions? Define or use a standard term.
We have changed the term as suggested
Expand the detail on the Spanish epidemiological situation to give the reader a better understanding. You note that the 5th wave saw a decrease in incidence, but then you note that there is “high prevalence of the disease in young age groups and pregnant women” – was this during the 5th wave only (why? Different variants?) or a feature of all waves?
We expanded the details of fifth covid 19pandemic wave in the introduction, as suggestred
Methods
Well described and seem appropriate and interesting. What about recall bias? You are asking participants to remember what their health was like two years ago before the first wave – how accurate/unbiased is this likely to be? You need a comment on this either here or in your limitations paragraph at the end of the discussion.
We have included the suggested comment about the recall bias as a limitation of our study.
Section 2.3 – why are you presenting results in the methods section? T1 and 2 and all accompanying text should be moved to the results section. All you need in the methods is a brief description of the population (perhaps with a short table showing the sociodemographic variables but not any of your clinical findings). Table 1: alignment in last column of first row (menstrual disorders) is out so hard to read correctly. It would make more sense to present first the sociodem variables (eg in the top half of the table) then the clinical variables – at present they are mixed up.
According to your suggestions we have remade Table 1. Now it includes only sociodemographic variables for the section Population. Clinical variables are shown as new Table 2 in the section Results
Results
This should start with tables 1 and 2 giving an overview of the respondents’ background characteristics and reported disorders/ill-health before and after the pandemic.
We have started the section as you suggested.
3.1: several times you use “re-experencing” – I think you mean “re-experiencing”
We have corrected it
Fig 1: define TEPT. Also explain E. I don’t understand why the multiple regression showed gender as the best predictor for PTSD but this doesn’t then appear in the best-fit SEM model – please explain.
We have changed TEPT (Spanish acronym) for PTSD (post-traumatic stress disorder). We have also revised SEM model for PTSD, and we have realized that there was a mistake in Figure 1, and the variable Gender was missing. We have corrected it.
Fig 2: this is not cited in your text – you need to refer to it and explain it. I don’t understand the presentation of the sequencing – you found that depression scores were higher in women who also reported having menstrual disorders, but the way you present the Fig makes it seem as though depression leads to menstrual disorders. This is not what you found. Reversing the arrow might help to clarify this.
We have revised the text and the model and we think that the wording was confusing. Our results do not show that having menstrual disorders is an independent variable acting as a depression risk factor. Rather, the analysis shows that depression is influencing the occurrence of menstrual disorders.
Discussion
Currently the first sections of this are rather thinly referenced with wider literature and you have many unsupported statements, which (however true they may be) need to be referenced. For example, your research does not examine the wider causes of the rise in mental health disorders that you found, therefore you need references to support statements like “The stress caused by the progress of the pandemic and the thousands of associated deaths, the absence of effective treatment, the saturation of hospitals, and long work shifts with potential exposure to infection have damaged the mental health of professionals” and “specialists have participated in virus containment strategies, proposing treatment protocols and measures to prevent the spread of the virus, on many occasions establishing strict isolation measures for patients” and “obstetricians and gynecologists in particular, have been working their usual shifts (when they were not overtime), with the additional responsibility of caring for school-age children at home in many cases, helping in their virtual schooling and, ultimately, supporting the family.”
As suggested, we have included some references to support these statements.
You could give more detail on the nature of the frontline work that is associated in your models with high mental health scores (across the various that you test for). What does frontline surgeon work in Spain involve, compared to delivery room specialists. Were there any differences between your male and female respondents in terms of where they worked?
We have included a paragraph to better clarify these concepts
Malaga 2021, December 10
Dear Editor of the Journal of Clinical Medicine:
We wish to thank the careful reading that the reviewers made of our article, as well as the comments and contributions, which undoubtedly have served to improve the quality of the study. We have done our best to incorporate the recommendations of the specialists who have reviewed the paper. We have considered point-by-point all the suggestions, including in the new document all the changes that are described below. We hope that the manuscript could be now accepted for publication in the journal.
On behalf of the authors,
Dr Jesús S Jiménez-López
Response to the Reviewers
REVIEWER 1
Abstract:
I suggest significantly reducing the first 2/3 (background and objectives) then extend the results a little and add a sentence or two on the important implications/conclusions you draw. In particular, I think the effect you find on women specialists, who have added burdens (eg home schooling children) is particularly important and you should reiterate your call for future pandemic responses to offer “appropriate psychological support” and ensure “human and material resources, flexible work shifts, and facilities to reconcile work and family life”
We have changed the abstract following your suggestions. We have highlighted the higher impact on women, and the call for future pandemic
Intro
What is meant by “non-delayed conditions”? Do you mean emergency conditions? Define or use a standard term.
We have changed the term as suggested
Expand the detail on the Spanish epidemiological situation to give the reader a better understanding. You note that the 5th wave saw a decrease in incidence, but then you note that there is “high prevalence of the disease in young age groups and pregnant women” – was this during the 5th wave only (why? Different variants?) or a feature of all waves?
We expanded the details of fifth covid 19pandemic wave in the introduction, as suggestred
Methods
Well described and seem appropriate and interesting. What about recall bias? You are asking participants to remember what their health was like two years ago before the first wave – how accurate/unbiased is this likely to be? You need a comment on this either here or in your limitations paragraph at the end of the discussion.
We have included the suggested comment about the recall bias as a limitation of our study.
Section 2.3 – why are you presenting results in the methods section? T1 and 2 and all accompanying text should be moved to the results section. All you need in the methods is a brief description of the population (perhaps with a short table showing the sociodemographic variables but not any of your clinical findings). Table 1: alignment in last column of first row (menstrual disorders) is out so hard to read correctly. It would make more sense to present first the sociodem variables (eg in the top half of the table) then the clinical variables – at present they are mixed up.
According to your suggestions we have remade Table 1. Now it includes only sociodemographic variables for the section Population. Clinical variables are shown as new Table 2 in the section Results
Results
This should start with tables 1 and 2 giving an overview of the respondents’ background characteristics and reported disorders/ill-health before and after the pandemic.
We have started the section as you suggested.
3.1: several times you use “re-experencing” – I think you mean “re-experiencing”
We have corrected it
Fig 1: define TEPT. Also explain E. I don’t understand why the multiple regression showed gender as the best predictor for PTSD but this doesn’t then appear in the best-fit SEM model – please explain.
We have changed TEPT (Spanish acronym) for PTSD (post-traumatic stress disorder). We have also revised SEM model for PTSD, and we have realized that there was a mistake in Figure 1, and the variable Gender was missing. We have corrected it.
Fig 2: this is not cited in your text – you need to refer to it and explain it. I don’t understand the presentation of the sequencing – you found that depression scores were higher in women who also reported having menstrual disorders, but the way you present the Fig makes it seem as though depression leads to menstrual disorders. This is not what you found. Reversing the arrow might help to clarify this.
We have revised the text and the model and we think that the wording was confusing. Our results do not show that having menstrual disorders is an independent variable acting as a depression risk factor. Rather, the analysis shows that depression is influencing the occurrence of menstrual disorders.
Discussion
Currently the first sections of this are rather thinly referenced with wider literature and you have many unsupported statements, which (however true they may be) need to be referenced. For example, your research does not examine the wider causes of the rise in mental health disorders that you found, therefore you need references to support statements like “The stress caused by the progress of the pandemic and the thousands of associated deaths, the absence of effective treatment, the saturation of hospitals, and long work shifts with potential exposure to infection have damaged the mental health of professionals” and “specialists have participated in virus containment strategies, proposing treatment protocols and measures to prevent the spread of the virus, on many occasions establishing strict isolation measures for patients” and “obstetricians and gynecologists in particular, have been working their usual shifts (when they were not overtime), with the additional responsibility of caring for school-age children at home in many cases, helping in their virtual schooling and, ultimately, supporting the family.”
As suggested, we have included some references to support these statements.
You could give more detail on the nature of the frontline work that is associated in your models with high mental health scores (across the various that you test for). What does frontline surgeon work in Spain involve, compared to delivery room specialists. Were there any differences between your male and female respondents in terms of where they worked?
We have included a paragraph to better clarify these concepts

Reviewer 2 Report
I would like to congratulate the authors for developing this interesting article. I recommend some minor revisions that might improve the paper. First of all, the introduction should include more relevant information on obstetrics-gynaecology and COVID-19. The current empirical background to explain the phenomena is not enough. I suggest enhancing the discussion with more scientific findings. Additionally, I recommend highlighting practical implications in the discussion and conclusions sections.
Author Response
Malaga 2021, December 10
Dear Editor of the Journal of Clinical Medicine:
We wish to thank the careful reading that the reviewers made of our article, as well as the comments and contributions, which undoubtedly have served to improve the quality of the study. We have done our best to incorporate the recommendations of the specialists who have reviewed the paper. We have considered point-by-point all the suggestions, including in the new document all the changes that are described below. We hope that the manuscript could be now accepted for publication in the journal.
On behalf of the authors,
Dr Jesús S Jiménez-López
REVIEWER 2
I would like to congratulate the authors for developing this interesting article. I recommend some minor revisions that might improve the paper. First of all, the introduction should include more relevant information on obstetrics-gynaecology and COVID-19. The current empirical background to explain the phenomena is not enough.
Thankyou very much for encouraging comment. We have included the required information in the introduction to contextualize the study.
I suggest enhancing the discussion with more scientific findings. Additionally, I recommend highlighting practical implications in the discussion and conclusions sections.
We have included some references in the discussion, and we have changed the wording in some paragraphs to add scientific findings and to highlight future implications, as you suggest.
